# *Trypanosoma cruzi* Isolates Naturally Adapted to Congenital Transmission Display a Unique Strategy of Transplacental Passage

Paula Faral-Tello,[a]* Gonzalo Greif,[a] Selva Romero,[b] Andrés Cabrera,[a,b,c] Cristina Oviedo,[b] Telma González,[b] Gabriela Libisch,[a] Ana Paula Arévalo,[d] Belén Varela,[e] José Manuel Verdes,[e] Martina Crispo,[d] Yester Basmadjián,[b] Carlos Robello[a,f]

[a]Laboratorio de Interacciones Hospedero Patógeno/UBM, Institut Pasteur de Montevideo, Montevideo, Uruguay
[b]Departamento de Parasitología y Micología, Facultad de Medicina, Universidad de la República, Montevideo, Uruguay
[c]Unidad de Microbiología, Instituto de Patobiología, Facultad de Veterinaria, Universidad de la República, Montevideo, Uruguay
[d]Laboratory Animal Biotechnology Unit, Institut Pasteur de Montevideo, Montevideo, Uruguay
[e]Unidad de Patología, Departamento de Patobiología, Facultad de Veterinaria, Universidad de la República, Montevideo, Uruguay
[f]Departamento de Bioquímica, Facultad de Medicina, Universidad de la República, Montevideo, Uruguay

**ABSTRACT** Chagas disease is mainly transmitted by vertical transmission (VT) in nonendemic areas and in endemic areas where vector control programs have been successful. For the present study, we isolated natural *Trypanosoma cruzi* strains vertically transmitted through three generations and proceeded to study their molecular mechanism of VT using mice. No parasitemia was detected in immunocompetent mice, but the parasites were able to induce an immune response and colonize different organs. VT experiments revealed that infection with different strains did not affect mating, pregnancy, or resorption, but despite low parasitemia, VT strains reached the placenta and resulted in higher vertical transmission rates than strains of either moderate or high virulence. While the virulent strain modulated more than 2,500 placental genes, VT strains modulated 150, and only 29 genes are shared between them. VT strains downregulated genes associated with cell division and replication and upregulated immunomodulatory genes, leading to anti-inflammatory responses and tolerance. The virulent strain stimulated a strong proinflammatory immune response, and this molecular footprint correlated with histopathological analyses. We describe a unique placental response regarding the passage of *T. cruzi* VT isolates across the maternal-fetal interphase, challenging the current knowledge derived mainly from studies of laboratory-adapted or highly virulent strains.

**IMPORTANCE** The main findings of this study are that we determined that there are *Trypanosoma cruzi* strains adapted to transplacental transmission and completely different from the commonly used laboratory reference strains. This implies a specific strategy for the vertical transmission of Chagas disease. It is impressive that the strains specialized for vertical transmission modify the gene expression of the placenta in a totally different way than the reference strains. In addition, we describe isolates of *T. cruzi* that cannot be transmitted transplacentally. Taken together, these results open up new insights into the molecular mechanisms of this insect vector-independent transmission form.

**KEYWORDS** Chagas disease, *Trypanosoma cruzi*, transcriptomics, transplacental, placental tropism, vertical transmission

Address correspondence to Carlos Robello, robello@pasteur.edu.uy.

*Present address: Paula Faral-Tello, Laboratory of Apicomplexan Biology, Institut Pasteur de Montevideo, Montevideo, Uruguay.

The authors declare no conflict of interest.

The acronym TORCH (toxoplasmosis, rubella, cytomegalovirus, and herpes simplex virus) refers to those pathogens that are transmitted in the human species from mother to offspring through transplacental passage. The acronym is more exclusive than inclusive, leaving out an expanding list of pathogens that are vertically transmitted and represent severe human health problems. In recent years, the relevance of this

transmission mechanism has become clear, and Chagas disease (CD) is a paradigmatic case. In the absence of the vector, *Trypanosoma cruzi* can still be transmitted through blood transfusions, organ transplants, or congenitally. Hence, the migration of asymptomatic individuals from endemic to nonendemic areas due to poverty conditions in the context of globalization has led to its current consideration as a reemerging disease (1). Unlike vectorial transmission, all other forms of transmission are possible in any country (2). The World Health Organization estimates that at least one-quarter of the total burden of newly reported cases corresponds to congenital Chagas disease (CCD) (3), which includes endemic areas with successful vectorial control programs and nonendemic areas where vertical transmission (VT) is the only active route of transmission. Although CD is considered a vector-borne disease, these events have radically changed its epidemiological profile, turning it into a global urban disease and a worldwide health problem (4, 5). The rates of *T. cruzi* vertical transmission range from 1% to 12%, depending on the area (6), with an average rate of 5% (7). CCD often presents as an acute infection, and although 60% of infected babies are born asymptomatic, they usually exhibit detectable levels of parasitemia and display higher frequencies of low weight, prematurity and lower APGAR scores (aspect, pulse, grimace, activity, and respiration) than noninfected babies (8–10). Although rarely lethal, untreated CCD can lead to hepatosplenomegaly, meningoencephalitis, and myocarditis (11); however, when detected and treated early, parasitic clearance is more than 90% effective (12, 13). One important aspect of CCD is that most infected pregnant women are diagnosed for CD during routine control screenings, because they are usually asymptomatic (7). This observation led us to the hypothesis that there could be natural strains adapted to vertical transmission, capable of persisting in a "silent" state in the host and able to cross the placental barrier during pregnancy. In addition, these microorganisms would constitute a successful case of parasitism and are not accurately modeled by highly virulent and laboratory-adapted strains. To test the hypothesis, we identified familial cases of congenital Chagas disease in which *T. cruzi* strains have asymptomatically passed through generations: from great-grandmothers, to grandmothers, and then to mothers and their children. By xenodiagnoses of babies born to positive mothers, we isolated several strains, and then we investigated them in terms of virulence and tissue tropism, as well as through the development of a vertical-transmission murine model coupled to the transcriptomic placental response.

## RESULTS

**Molecular typing of isolates.** Xenodiagnostic tests were performed on three newborns whose mothers and grandmothers were seropositive for CD, and after 15 to 25 days, parasites were observed in insect feces. Insect intestinal homogenates were inoculated into BALB/cJ mice, and parasites were detected in blood samples after 30 to 45 days, although in all cases, they were observed after counts of at least 100 microscopic fields. Parasitemia peaks were obtained only after immunosuppression, and three isolates were obtained and named TcLu (case 1), TcKr (case 2), and TcGi (case 3). Thereafter, due to their very low virulence in immunocompetent mice, all of the isolates were maintained by passages in Nu/J mice, which do not control the infection. Molecular typing indicated that TcGi, TcKr, and TcLu belonged to the discrete typing unit (DTU) TcV or BCb lineage (Fig. S1 in the supplemental material) (14).

**Murine infection assays indicate that VT isolates reproduce their clinical features in the murine model.** To compare the infection course and virulence of the different strains, Kaplan-Meier and parasitemia curves with different parasite loads were performed on BALB/cJ mice. As shown in the survival curves (Fig. 1A), 100% survival was observed both for mice infected with strain Dm28c (DTU TcI strain with moderate virulence [MV]) and for mice infected with the VT strains when $5 \times 10^4$ trypomastigotes were inoculated. On the other hand, when inoculated with $1 \times 10^4$ parasites of the Garbani strain (DTU TcVI strain with high virulence [HV]), 100% of the mouse population reached the endpoint by approximately day 19 postinfection (p.i.), and with an inoculum of $1 \times 10^3$ trypomastigotes, 50% of the mice reached the endpoint by day 29. As shown in Fig. 1B, an initial inoculum of $5 \times 10^4$ parasites of the Dm28c strain

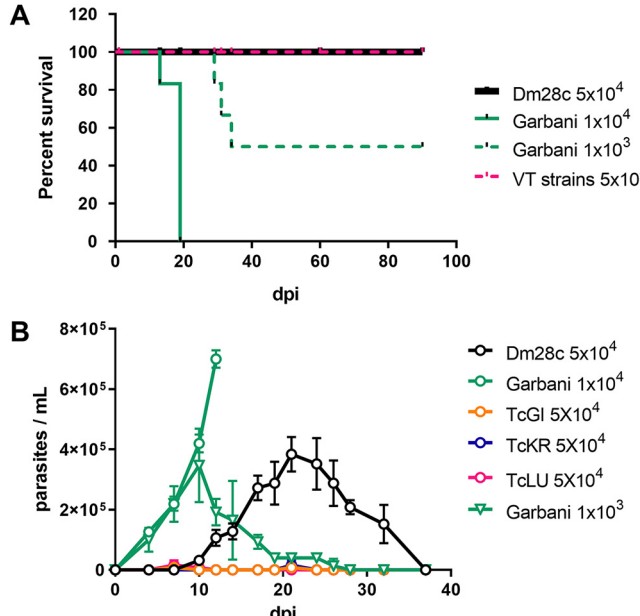

**FIG 1** (A) Survival curves of mice infected with Dm28c (MV), Garbani (HV), and VT strains with the indicated initial inoculums. (B) Parasitemia along the course of infection of the different strains with the indicated inoculums. Error bars show standard errors.

generated a peak of parasitemia of $4 \times 10^5$ parasites per mL at day 20 p.i., and by the 36th day, no parasites were observed. Initial inoculums of $1 \times 10^4$ of the Garbani strain exhibited a peak of parasitemia of $7 \times 10^5$ parasites/mL on day 12 p.i., from which the animals did not recover (Fig. 1B). When the Garbani inoculum was reduced to $1 \times 10^3$, 50% of the surviving mice showed a parasitemia curve with a peak of $4 \times 10^5$ parasites per mL on day 10 p.i., and parasite clearance from blood was observed on day 28. However, for the VT strains, parasitemia was almost undetectable (Fig. 1B) when $5 \times 10^4$ parasites were inoculated. These results led us to the conclusion that the VT strains present low virulence and a silent phenotype in mice, whereas the Garbani and Dm28c clones exhibit high and moderate virulence, respectively, which is why they were used for comparative studies throughout this work.

**Blood trypomastigotes of congenital Chagas isolates are unable to perform epimastigogenesis *in vitro*.** Epimastigogenesis capacity was evaluated according to Kessler et al. (15). After stress, a day-to-day follow-up was conducted to count epimastigotes and intermediate forms according to flagellar form, length, and nucleus/kinetoplast position. As shown by the results in Fig. 2A and B, at the start of the experiment, mostly trypomastigotes were observed and no intermediate forms were counted. By the 6th day, approximately 60% of the populations of the Dm28c and Garbani strains corresponded to epimastigote forms and 40% to intermediate forms, while the total populations of parasites from VT isolates (TcGi, TcKr, and TcLu) corresponded to intermediate forms. Finally, stable 100%-replicating epimastigote populations were observed for the Dm28c and Garbani strains on the 10th day of incubation, but a different scenario of arrested intermediate forms was observed for the VT strains (Fig. 2A and B, TcGi, days 2, 6, and 10). It is important to mention that differentiated Dm28c and Garbani epimastigotes were able to replicate and maintain a stable replicating population, whereas the cultures of VT strains (intermediate forms exclusively) were not able to replicate and died around the 20th day. Attempts to maintain them in culture by changing the medium and adding supplements failed to induce replication or rescue the parasite population.

**VT isolates can colonize different organs and induce a systemic immune response similar to high and moderate virulence strains.** At day 30 p.i., parasite DNA was detected in all organs regardless of the strain, and no statistically significant differences were found (Fig. 3A). In gut tissues, similar loads were observed for all the strains, but

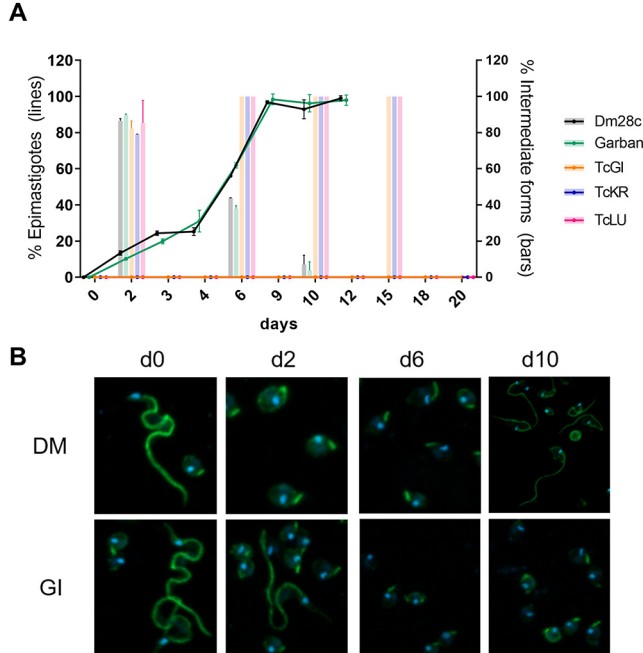

**FIG 2** (A) Proportions of epimastigotes (lines) and intermediate forms (bars) out of total population of trypomastigotes during the epimastigogenesis experiment that lasted 20 days. Error bars show standard errors. (B) Immunofluorescence analysis of the epimastigogenesis process of two strains, Dm28c and the VT strain TcGi, which did not differentiate over time.

with higher values than those recorded for the rest of the tissues, suggesting that all the strains used in this particular model shared a preferential tropism. As expected, when we observed mean values, parasite loads diminished almost to the noninfected-control level in all tissues and for all strains after day 60 p.i. (Fig. 3B, dotted line at a value of 1). Dm28c-infected organs like the cardiac muscle, gut, and uterus showed positive mean values, but differences from the results for the other strains were not significant. Notably, the parasite load of the TcKr strain in the uterus did not change within 30 or 60 days p.i.

The splenic response at day 30 p.i. measured by mRNA expression showed strong upregulation of different cytokines (gamma interferon [IFN-$\gamma$], interleukin-12 [IL-12], tumor necrosis factor alpha [TNF-$\alpha$], IL-6, IL-10, IL-4, and transforming growth factor-$\beta$ [TGF-$\beta$]) induced by infection with all parasite strains, as shown by the results in Fig. 3C. All the cytokines evaluated were overexpressed in infected mice compared to their expression in uninfected control animals (Fig. 3C, gray bars). It is noteworthy that the expression levels of each cytokine were comparable among all the strains, regardless of the differences in virulence described above. Garbani showed higher levels of expression for most genes than Dm28c, except for the anti-inflammatory cytokine IL-4. A strong and more likely proinflammatory response was observed for the TcGi and TcLu strains, whereas TcKr induced lower levels of expression, except for TGF-$\beta$. These results indicate that despite their low virulence, the VT strains reached the different organs and induced an immune response similar to that of the high and moderate virulence strains. At 60 days p.i. (Fig. 3D), cytokine expression in infected groups fell to control levels, compatible with an ending of the acute phase, with the exception of the IL-12, which remained significantly upregulated in mice infected with the VT strains compared to the control group, suggesting a long-lasting protective response.

**Infections with the different strains do not significantly affect reproductive variables.** To address whether mating, pregnancy, and resorption rates were altered with infection and whether the strains were vertically transmitted to the offspring, a vertical transmission model was optimized (Fig. 4A). As shown by the results in Fig. 4B,

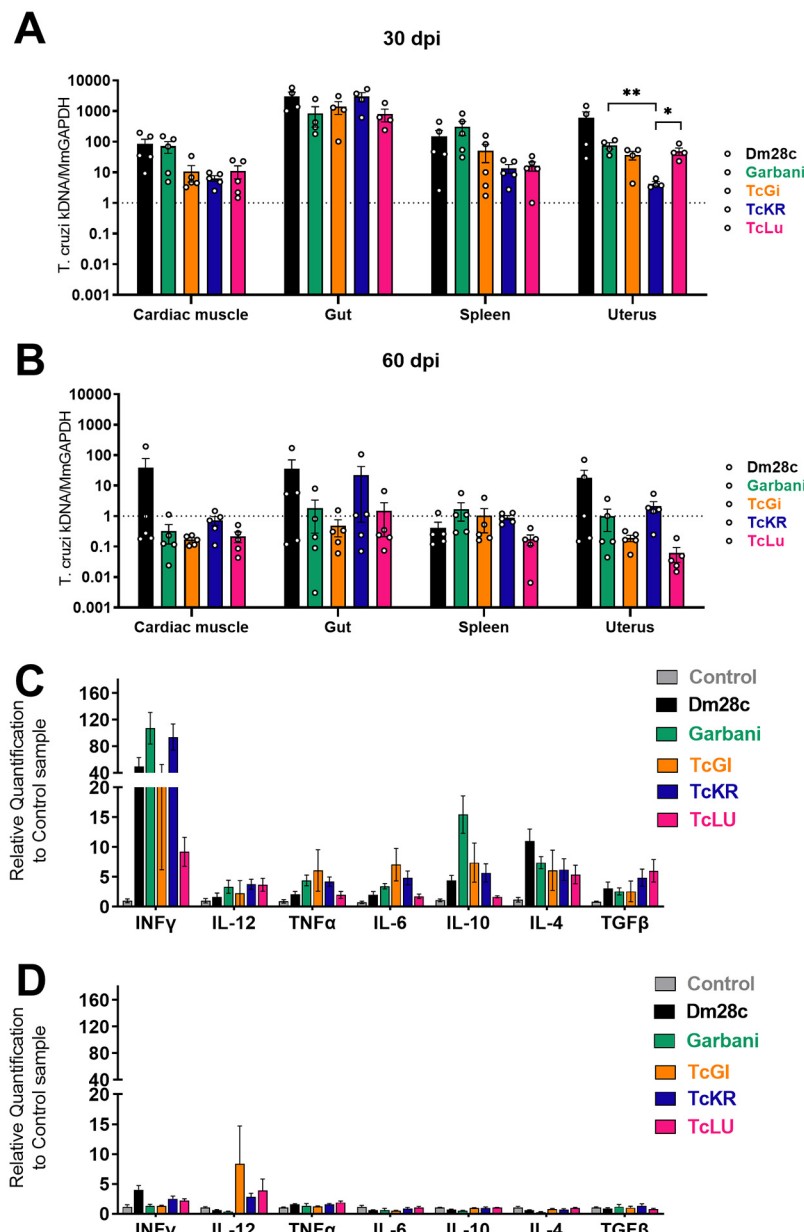

**FIG 3** (A, B) Parasite loads in different tissues of mice infected with the different strains of *Trypanosoma cruzi* at 30 dpi (A) and 60 dpi (B). Mm, *Mus musculus*. (C, D) Levels of expression of cytokines in splenic cells at 30 dpi (C) and 60 dpi (D). Error bars show standard errors. *, $P < 0.05$; **, $P < 0.01$.

mating rates decreased with infection; while the mating rate of the control group was around 50%, infected groups had significantly lower rates, 40% for the group infected with the Garbani strain and 20% for those infected with Dm28c, TcGi, TcKr, and TcLu. Regarding pregnancy rates and resorption rates (measurement of implantation fate), no significant differences were observed for any of the groups compared to the control group. Altogether, these results indicate that under our conditions, neither infection nor the strain had a negative impact on fertility or fetus viability.

**VT strains are more efficient than reference strains in transplacental transmission.** After delivery, fetal, placental, uterine, and resorption tissues were processed to detect parasite DNA. For each sample, an equivalent amount of tissue from noninfected mice was processed to set up a negativity threshold, which was set at a value of 1 (Fig. 4C). A great heterogeneity was found among parasite loads in placental and fetal tissues, as shown by the results in Fig. 4C. In the groups infected with the VT

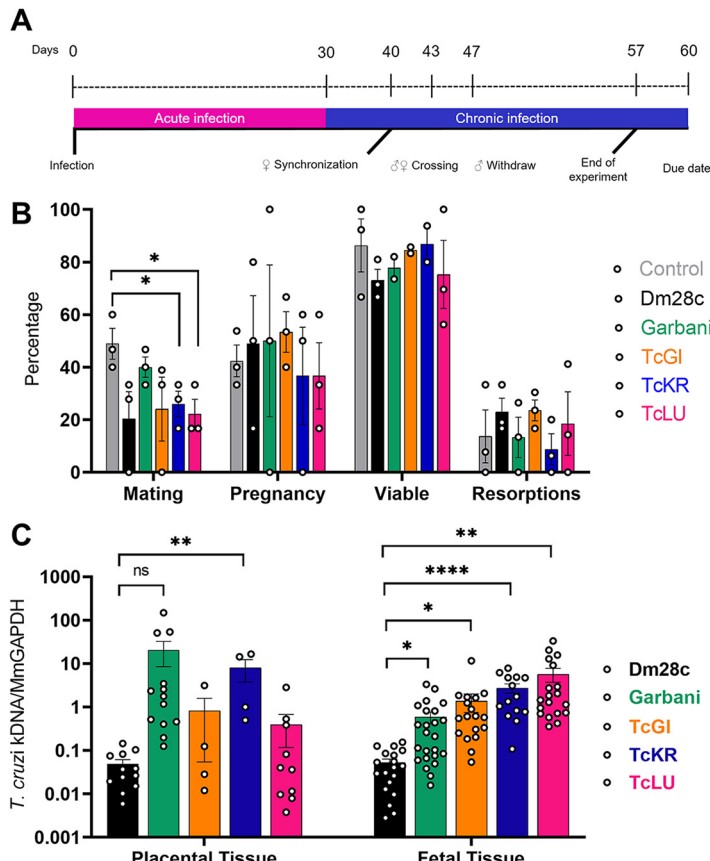

**FIG 4** (A) Vertical transmission experiment scheme. (B) Rates of mating, pregnancy, viable fetuses, and resorptions obtained from the vertical transmission experiment for the different experimental groups. (C) Parasite loads determined in placental and fetal tissue samples for the different experimental groups. Error bars show standard errors. *, $P < 0.05$; **, $P < 0.01$; ****, $P < 0.0001$; ns, not significant.

strains, no parasite loads were found in the uterine tissue or the placenta (except for TcKr), and remarkably, all the groups infected with the VT strains showed significantly higher parasite loads in the fetal tissues, indicating that, regardless of low virulence and parasite absence in the placenta, parasites were able to pass through the organ into the fetus more efficiently than the other (more-virulent) strains, probably indicating transplacental passage early in gestation.

**VT strains are more infective in human trophoblast-derived cells than in fibroblasts.** After 2 h of incubation, all the strains were able to infect *in vitro* both human fibroblast and trophoblast cell lines. When using laboratory reference strains, the human foreskin fibroblast (HFF) line was more susceptible to infection by *T. cruzi* than the Swan 71 trophoblast cells, whereas the VT strains showed low infectivity in both cell lines; however, all of them infected Swan cells more efficiently (Fig. 5A). This could be better visualized by determining the Swan/HFF infection indexes: while the VT strains showed relative values of 4 and 10, the values for the Garbani and Dm28c strains showed an inverted tendency, with values below 1 (Fig. 5B).

**VT isolates induce a unique pattern of placental gene expression.** The effects of the different strains of *T. cruzi* on placentas were studied at the gene expression level by high throughput RNA sequencing. Taking into account the high variability of *in vivo* experiments, we carried out a principal component analysis (PCA), wherein three groups were clearly identified (Fig. 6A): (i) the Garbani strain, (ii) the Dm28c strain and noninfected mice, and (iii) VT isolates, with the strains corresponding to high, moderate, and low virulence, respectively. Their correlation in the hierarchical clustering of the differentially expressed genes (DEG) is shown in Fig. 6C. When a cutoff of $-1.5 \geq \mathrm{Log}_2\mathrm{FC} \geq 1.5$

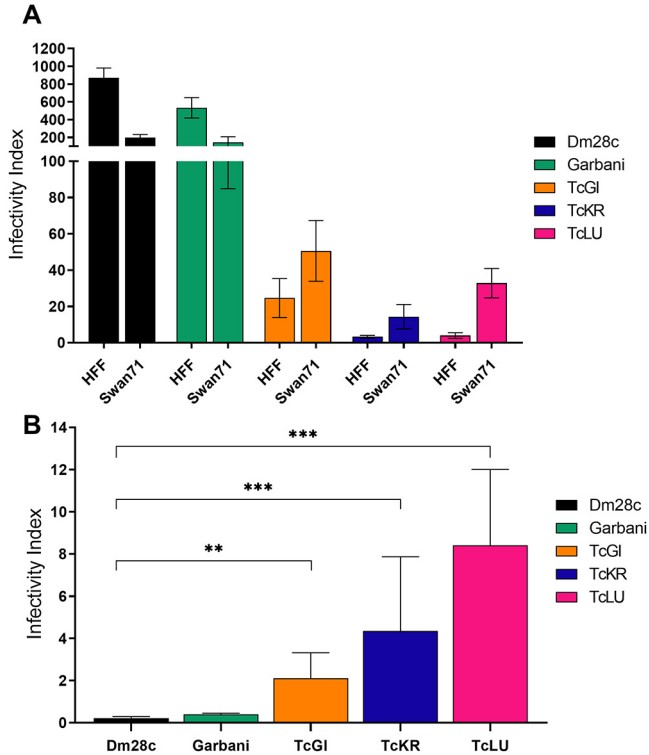

**FIG 5** (A) Infectivity index determined for each *T. cruzi* strain in two different human cell lines, HFF (fibroblasts) and Swan 71 (trophoblasts). (B) The relationships between the Swan 71 infection index and the HFF infection index determined for the different strains. Error bars show standard errors. **, $P < 0.01$; ***, $P < 0.001$.

(where FC is fold change) and an adjusted *p*-value of <0.01 were considered, the highly virulent Garbani strain modulated the expression of 2,507 genes (1,114 up and 1,393 down compared to the uninfected control condition), the VT strains modulated 408 genes (232 up and 176 down), and only two overexpressed genes were found in placentas of mice infected with Dm28c (Fig. 6C and Table S3). The relationship between up- and downregulated genes was close to 1 for all conditions; none of the differentially expressed genes was shared by all the strains, and only 29 genes were shared by the Garbani and VT strains (8 up and 21 down) (Fig. 6B and Table S4). When the 30 most expressed genes (Fig. S2) or the most significant differentially expressed genes (Fig. S3) were compared between strains, their respective heatmaps were opposites; in other words, placental responses to high- and low-virulence vertically transmitted strains were diametrically opposed. The same opposite images were obtained when Gene Ontology (GO) was analyzed (Fig. 7). Notably, the most significant GO terms in response to the highly virulent strain were inflammation, cellular immune response, and ribosomal proteins, all of them at the expense of overexpressed genes, while the most significant terms in response to the VT strains (mitosis, meiosis, cell cycle, DNA replication, and response to DNA damage) consisted mainly of downregulated genes. In addition, the processes enriched in one condition were not affected in the other. When we dissected each term into its components, a surprising number of up- or downregulated genes were found as an exclusive response to each condition (Fig. S4). On the other hand, terms that were significantly downregulated in Garbani-infected mice were related to transport, lipid metabolism, cell morphology, and adherence, whereas the upregulated terms in mice infected with the VT strains were import- and secretion-related genes, which were downregulated in the placentas of mice infected with the highly virulent strain.

Specific gene families exhibiting inverse regulation in the Garbani- and VT strain-infected mouse placentas are shown in Fig. 8. Among these groups, those worth mentioning are the

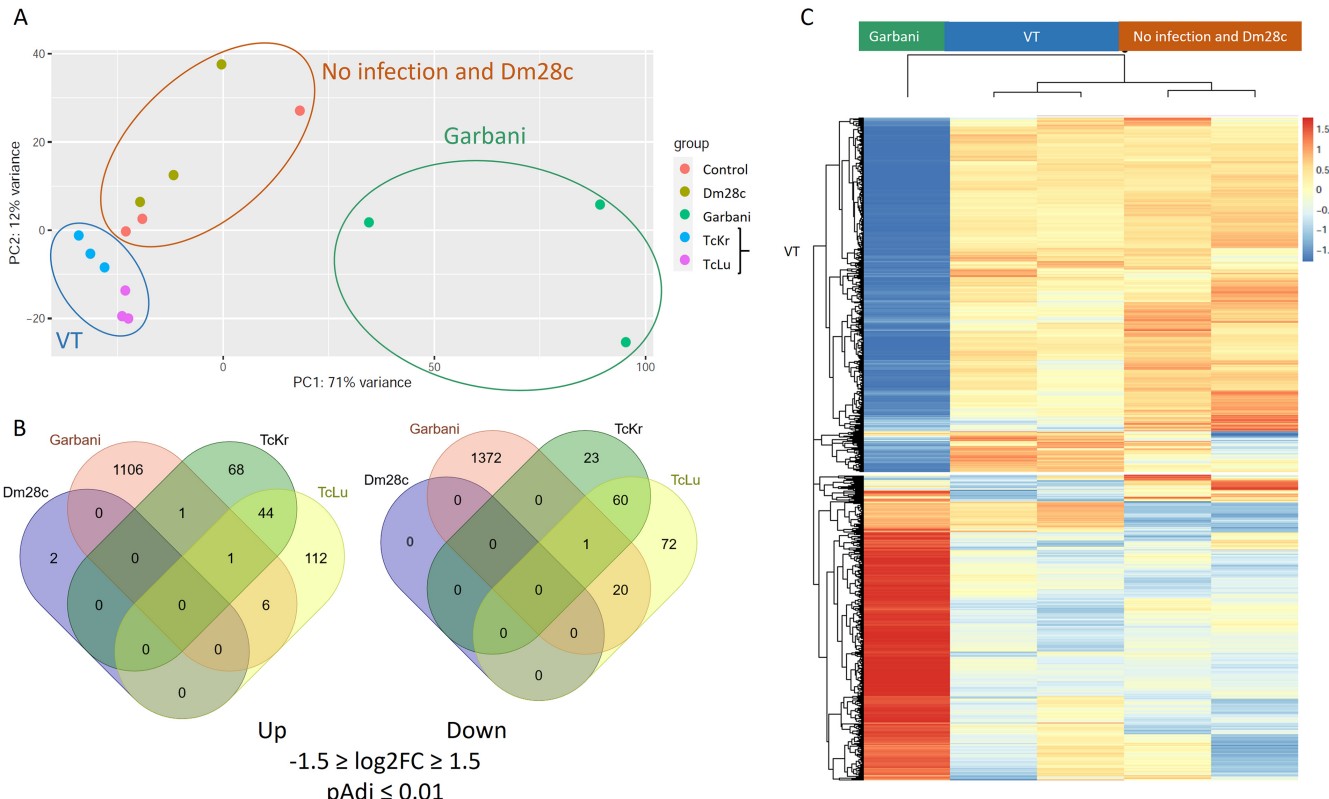

**FIG 6** (A) Principal component analysis (PCA) of RNA-seq data of the different biological replicates of placentas. (B) Venn diagrams of differentially expressed genes (DEGs) (1.5 ≥ Log$_2$FC ≥ 1.5 and $P_{adj}$ ≤ 0.01 [Benjamini-Hochberg-adjusted $P$ value]) in infected placentas compared to control placentas. (C) Hierarchical cluster analysis of normalized read counts of DEGs. No infection, control mouse placentas; Garbani, high virulence (HV) strain; Dm28c, medium virulence (MV) strain; TcKr and TcLu, VT strains.

ones playing important roles in the maternal-fetal interphase: immunomodulators, such as genes encoding pregnancy-specific glycoproteins (*psg*), carcinoembryonic antigen cell adhesion molecules (*ceacam*), and prolactins (*prl*), in addition to those related to cell permeability and cell adhesion, such as solute carriers (*slc*), sodium channels (*scn*), desmogleins (*dsg*), and claudins (*cld*). Differential regulation among the Garbani and VT isolates also involved matrix metalloproteinases (*mmp*) and the adamlysin family (*adam*) and their intracellular (*timp*) and secreted (*sfrp*) inhibitors. Gene families involved in defense mechanisms showed opposite modulation under the different conditions; for instance, metallothioneins (*Mt*), proteins that sequester metals, were upregulated in the Garbani group, while signaling lymphocyte activation molecules (*slam*) were inflammasome-related factors.

In summary, the three groups found concerning *T. cruzi* strain and placental relation/ tropism also had differential placental responses evidenced in the RNA-seq experiment: while placentas of the group infected with the moderate-virulence strain resembled control placentas (and in fact, transmission did not occur), the high- and low-virulence groups exhibited completely different and opposite responses.

**The molecular footprint of placentas correlates with histopathological alterations.** Results from the analysis of placental sections stained with hematoxylin and eosin (H&E) indicated that the placentas of the Garbani group were the most affected, showing high scores of edema, inflammation, necrosis, and tissue degeneration. As shown by the images in Fig. 9, pathological alterations were found in both the maternal decidua basalis and the labyrinth zone of placentas of mice infected with the Garbani strain. Regarding edema and inflammation scores, the Garbani strain topped the ranking, followed by Dm28c, and last, TcLu, with mild inflammation. TcKr placentas did not show inflammation, degeneration, or necrosis. It is important to mention that no amastigote nests were identified in any of the samples using this staining method.

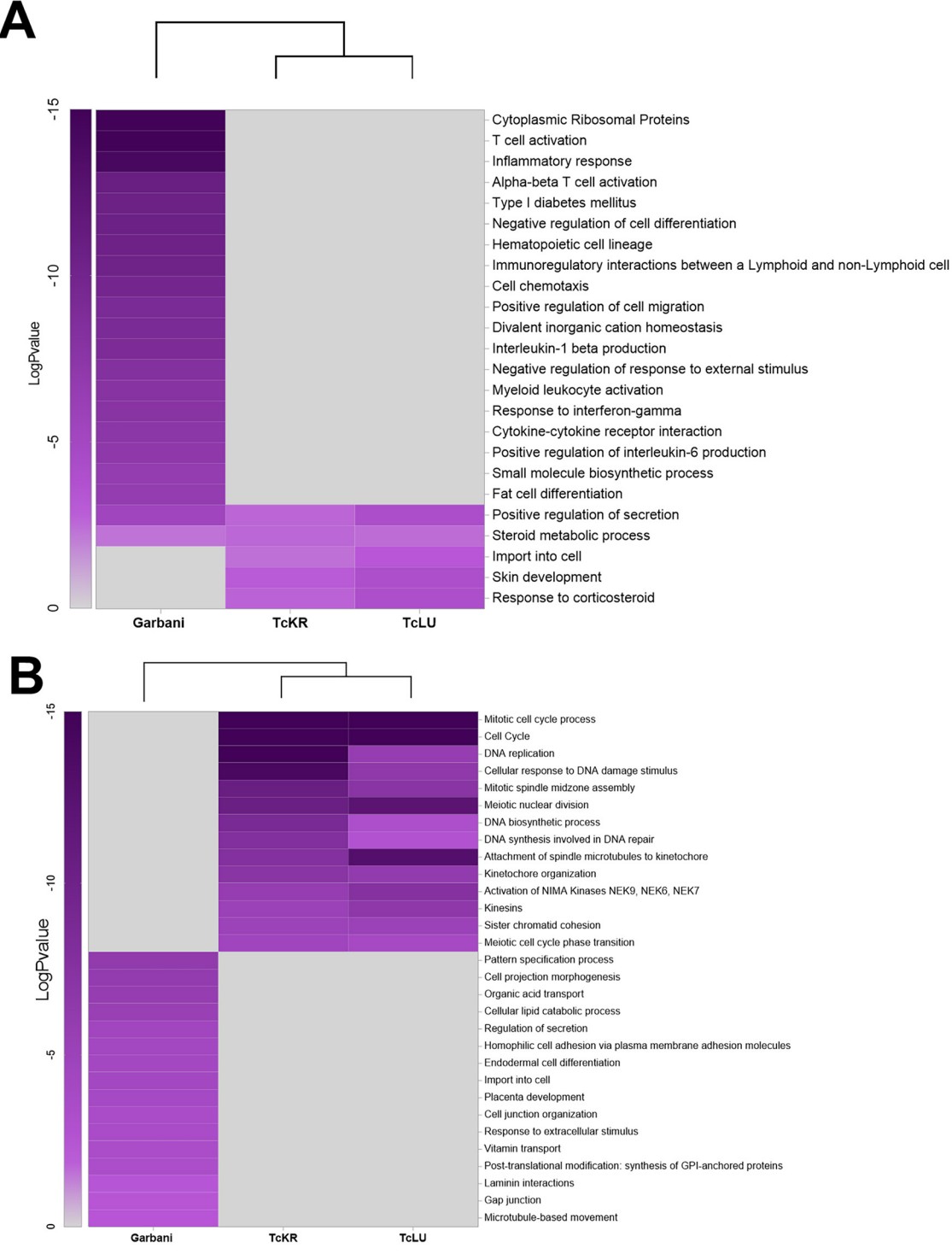

**FIG 7** Heatmaps showing enriched GO terms among the upregulated DEGs (A) and downregulated DEGs (B) in placentas derived from mice infected with HV strain Garbani or VT strain TcKr or TcLu.

## DISCUSSION

This work began with clinical observation. In the majority of congenital Chagas disease cases, the mothers are diagnosed during pregnancy at routine control screenings, due to the absence of symptoms, and familial clustering (several cases of congenital Chagas disease among one family cluster) is frequently observed (7, 12, 16). In this context, our working hypothesis was that there were *T. cruzi* strains adapted and specialized

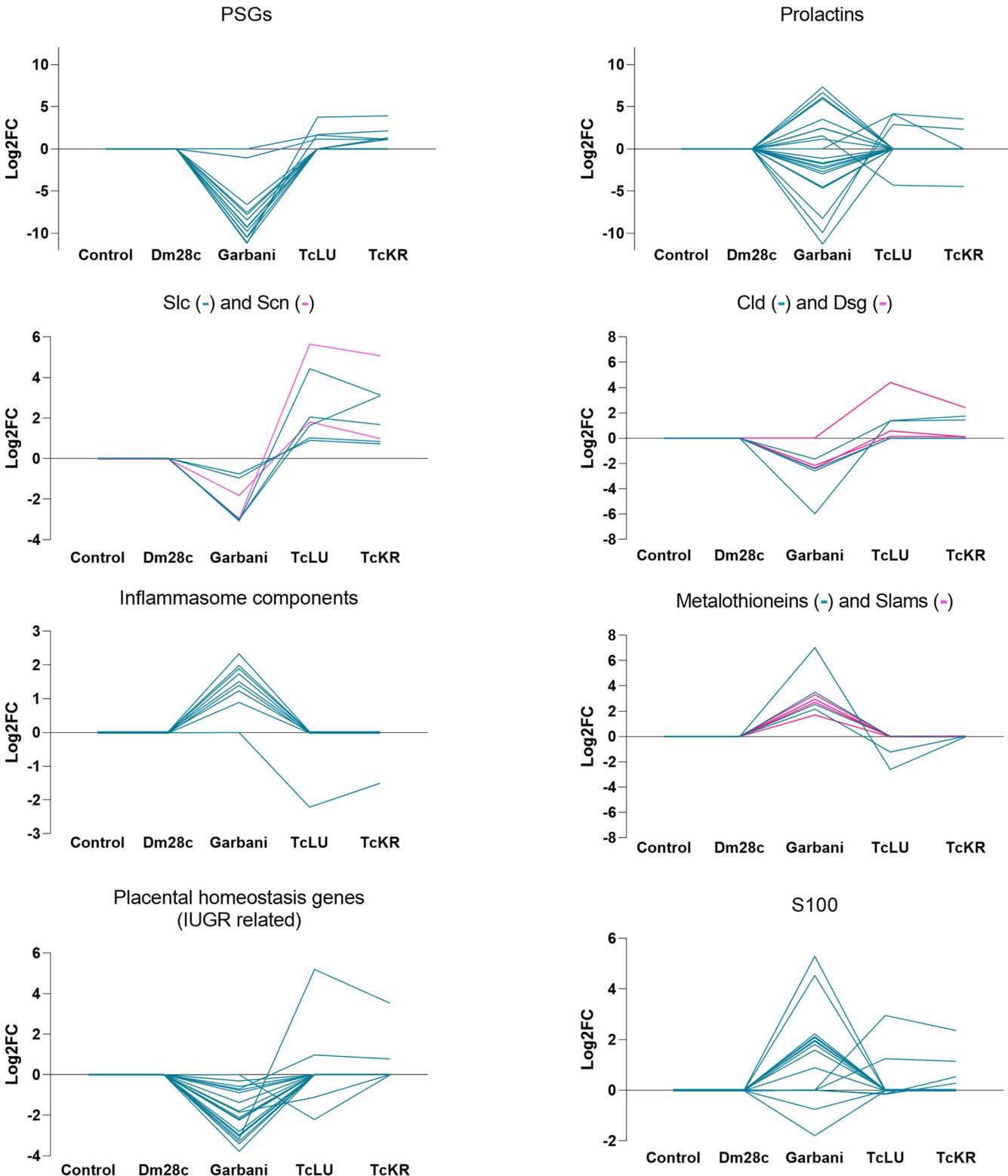

**FIG 8** Differential expression (log$_2$FC) found in groups of genes of interest compared between the different experimental groups.

to vertical transmission that probably differed substantially from commonly used laboratory strains. We selected the particular scenario of a formerly endemic region where vectorial transmission has been uninterrupted for at least 25 years (17). We focused on babies born under the following conditions: (i) Chagas disease detected in the mother

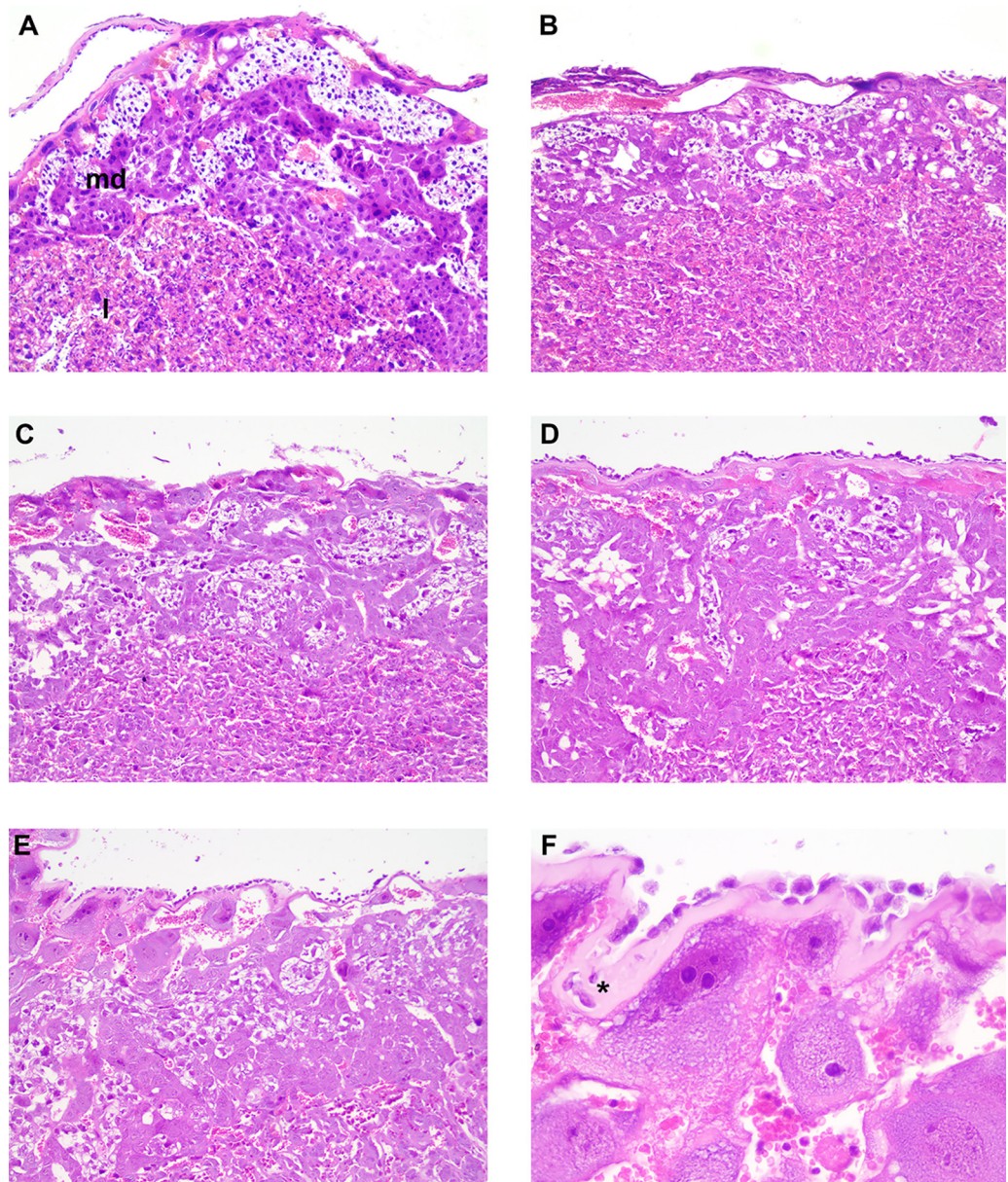

**FIG 9** Histopathological analysis of placentas. Note that in all images, the maternal side is at the top and the fetal side is at the bottom. (A) Maternal decidua basalis (md) and fetal labyrinth (l) are shown (control, H&E, ×10 magnification). (B) Mild edema of decidua basalis (Dm28c, H&E, ×10 magnification). (C) Mild edema of decidua basalis (TcLu, H&E, ×10 magnification). (D) Moderate edema of decidua basalis and labyrinth (TcKr, H&E, ×10 magnification). (E) Moderate edema of decidua basalis and labyrinth (TcGa, H&E, ×10 magnification). (F) Detail of moderate edema of decidua basalis (asterisk) (TcGa, H&E, ×40 magnification).

during routine pregnancy studies, (ii) asymptomatic mother, (iii) familial clustering, and (iv) *T. cruzi* vertically transmitted to the baby, from which we obtained isolates. Throughout this work, these isolates are generically termed VT isolates (for vertical transmission), whereas the reference strains of high (Garbani) and moderate (Dm28c) virulence are referred to below as HV and MV strains, respectively.

The first question was whether mice infected with VT isolates would mimic human-like clinical behavior and outcome, i.e., low virulence, undetectable parasitemias, and classical *T. cruzi* persistence strategies. Both the survival curves and the persistence tests confirmed that they had very low virulence. Likewise, we determined in the *in vivo* assays that Garbani and Dm28c were strains of high and moderate virulence,

which made them suitable controls (Fig. 1). VT strains, however, showed a capacity for persistence, since they were able to infect all organs studied (heart, gut, spleen, and uterus) (Fig. 3) and induce a systemic immune response comparable to that observed during infection with the HV and MV strains. *T. cruzi* infection in BALB/cJ mice induced a Th1 response (18), and we found that in all cases, IFN-$\gamma$, IL-12, TNF-$\alpha$, and IL-6 were upregulated, as well as the anti-inflammatory cytokines like IL-10 and IL-4, without significant differences between strains. Both parasite clearance from organs and control levels of cytokine expression were reached by day 60 p.i. in mice belonging to all infected experimental groups (Fig. 3). These results clearly show that VT parasites harbor intrinsic characteristics that allow them to disseminate and induce a competent immune response but are virtually nondetectable in blood. We can assert that the murine model reproduces the characteristics of low virulence observed in humans and that we have two adequate controls (moderate- and high-virulence strains) for subsequent comparative studies.

The second question was to evaluate the impact of VT strain infection on reproductive and vertical transmission parameters (Fig. 4). We developed a murine model of mating and pregnancy in mice and found that *T. cruzi* infection affected mating in all cases, regardless of the responsible strain, although not due to virulence, since the HV strain showed the highest mating percentage. On the other hand, the infection did not affect pregnancy or spontaneous abortion rates, since the differences observed in these parameters were not significant. It is worth mentioning that time of infection is an important factor for pregnancy outcome in female mice, since going through the acute phase shortly after embryo implantation induces unhealthy pregnancy outcomes (19, 20). Despite the effect on mating, we did not observe any impact on pregnancy rates in our model, and our results are in accord with other reports (21, 22). The finding that parasitic loads were higher in fetal tissues of VT strain-infected groups than in fetal tissues of reference infected groups indicated that VT strains were specialized in vertical transmission (Fig. 4C). Moreover, VT strains exhibited a more efficient persistence strategy, since they colonized different organs and induced an immune response without generating major damage in the host (this was demonstrated in our murine model, and clinical data indicates that this is also valid in humans). The VT strain transmission mode constitutes an example of highly successful parasitism which additionally dispenses of the insect vector. Concerning the reference strains, there were two completely different situations: (i) the MV strain-infected group showed the highest parasitic load in the uterus and the lowest signal in the placenta and the parasites were not transmitted vertically at all, while (ii) parasites in the HV strain-infected group were vertically transmitted, although with lower efficiency (low signal in fetal tissue and fewer infected individuals among the offspring) than the VT strains, and showed a low parasitic load in the uterus and a positive signal in the placenta. The inability of the MV strain to transmit vertically is a phenomenon that requires attention. Although it is beyond the scope of this work, we want to point out that this finding opens a question for future investigation: what is present or missing in the Dm28c strain that prevents it from being vertically transmitted? On the other hand, the HV strain's capacity for vertical transmission leads us to postulate that it is due to high virulence at the cost of infecting almost any tissue and organ, which does not imply a specialized mechanism or placental tropism. In addition to being an example of inefficient parasitism, this type of strain, widely used in laboratories, does not seem to resemble the behavior expected for circulating strains in humans. Moreover, the observed placental tropism of VT strains has a correlation in an *in vitro* human model: a comparison of the Swan (trophoblast-derived) index versus the HFF index can be used to show that this relationship was inverted in VT strains. The Swan/HFF infection index was higher in the VT strains than in the MV and HV strains (Fig. 5). Although there is a difference between mouse and human placentas regarding the depth of trophoblast invasion into the decidual tissue, both are hemochorial placentas in which the fetal trophoblast is in direct contact with maternal blood. Finally, to further explore *T. cruzi*'s capacity for adaptation to the mammal side of the cell cycle (and why not the human side), we assayed *in vitro* epimastigogenesis. As

suspected, whereas the reference strains shifted toward epimastigotes in the expected period, VT strains remained in intermediate form and never replicated (Fig. 2). This process, which occurs naturally in the digestive tract of triatomine vectors, can be reproduced both *in vivo* (23) and *in vitro* (15, 24). Given that *in vivo* epimastigogenesis is only partially modeled by the *in vitro* process and that VT strains were able to replicate in the triatomines from which they were isolated, our results suggest that parasite plasticity allows specialization and adaptation to nonvectorial transmission. Similar observations were made in other trypanosomatids, such as in African trypanosomes that were mechanically transmitted between cattle during their introduction to America (in the absence of their natural vector, the tsetse fly) and had selection occur toward parasites specialized in the mammalian cycle, including the loss of guide RNAs necessary for complete mitochondrial-gene editing (25). Future analysis of comparative genomics and gene profiling of VT strains is needed to dissect the parasite molecular mechanisms involved in the adaptation to human infection through vertical transmission.

Our next question was, are there changes at the placental gene expression level that could explain the specialization of VT strains for vertical transmission? We performed RNA-seq on the placentas of mice in groups infected with the different strains, and the main results can be observed in a PCA graph, showing the presence of three groups (Fig. 6A). Hierarchical clustering of differentially expressed genes (Fig. 6C) confirmed that HV and VT strains induced totally different placental responses. First, out of approximately 3,000 differentially expressed genes, most of them (~87%) corresponded to placentas of the HV strain-infected group and ~13% to placentas of VT strain-infected groups, and only 2 genes were found upregulated in the Dm28c (MV) strain-infected group. Remarkably, none of the differentially expressed genes was shared by all strains, and only 29 genes were shared by the HV and VT strains. On one hand, the MV strain was unable to vertically transmit at all. Among those strains that were transmitted, there were two scenarios: while the HV strain induced dramatic changes in placental gene expression, mainly involving upregulation of genes related to inflammation, the VT strains did not affect immune response-related genes but did downregulate genes belonging to mitosis, meiosis, and cell cycle processes, probably inducing an arrested state in trophoblasts and other cell types present in the placenta. The HV strain-infected group's most significant changes involved gene induction and GO terms related to immune response, inflammatory processes, and ribosomal proteins (see below). A gene-by-gene heat map overview showed that the placental responses resembled opposite images, where genes upregulated in the placentas of Garbani (HV) strain-infected mice were downregulated or unchanged in the placentas of VT strain-infected mice and vice versa.

As mentioned, one of the main differences among placental responses to infection with the highly virulent strain and VT strains was in the immune and inflammatory processes (Fig. S4). In addition, analysis of upregulated genes belonging to the immune response GO term in VT strains showed that all were related to tolerance and anti-inflammatory processes, being unchanged or downregulated in HV strain-infected placentas (Fig. S4, blue frame). Among them, it is worth mentioning *s100a14*, a gene that was found to be downregulated in the placentas of Chagas-seropositive mothers (26). The protein encoded by this gene is known to induce the *mmp2* gene (matrix metalloproteinase 2) (27, 28), which we found did not change under any conditions and which is capable of inducing macrophage migration (29). Its upregulation has also been linked to a condition of placental tissue inflammation called histologic chorioamnionitis (30). Some of the S100 proteins function via the "nutritional immunity" mechanism by sequestration of essential metals required for intracellular parasites to thrive (31). Remarkably, eight other *s100* genes that react to damage-associated molecular patterns (DAMPs) (*s100a1*, *s100a3*, *s100a4*, *s100a6*, *s100a7*, *s100a9*, *s100a10*, and *s100a11*) were upregulated in HV strain-infected placentas and not in VT strain-infected placentas. The fact that these eight genes are receptor for advanced glycation end products (RAGE) dependent is a fundamental difference from *s100a14* (downregulated in HV strain infection), whose activation is RAGE

independent. Since RAGE is a proinflammatory pathway, the activation of RAGE-independent defense mechanisms may be attenuated in an anti-inflammatory environment like that found in VT strain-infected placentas.

Metallothionein genes *mt1*, *mt2*, *mt3*, and *mt4* were upregulated in the HV group and downregulated or unchanged in VT groups (Fig. 8). Their gene products are cytosolic proteins that mitigate metal poisoning and alleviate superoxide stress (32). Inflammasome-associated cytokines have recently been described to be constitutively released from placentas in healthy pregnancies. This state is considered an immunological signature of health, which is far from the past belief of an immunosuppressive state (33). We found that major inflammasome components were differentially regulated in the HV strain-infected placentas, which suggested an unhealthy state that did not occur in VT strain-infected placentas. Inflammasome components *nlrp1*, *nlrp3*, *pycard*, *gsdmd*, *casp1*, *casp4*, and *casp12* were upregulated in the placentas of the HV strain-infected group and unchanged in the VT strain-infected group (Fig. 8). Upregulation of inflammasome genes during placental infection with *Listeria monocytogenes* (a bacterium associated with poor pregnancy outcomes) has been described (34), similar to what we found for the HV strain but not in VT strain-infected placentas.

The pregnancy-related genes (pregnancy-specific glycoproteins [PSGs]) with important roles in the maternal-fetal interphase were found to be differentially modulated between the two transplacental-passage models. PSGs are the most abundant trophoblastic proteins in maternal blood during pregnancy. They have immunomodulatory, proangiogenic and antiplatelet functions and regulate maternal-fetal interactions (35). Low levels of PSGs (observed in HV but not in VT strain-infected placentas) are associated with fetal growth restrictions (35). The fact that 16 PSGs were downregulated in placentas infected with the HV strain and unchanged in VT strain-infected placentas (Fig. 8) correlates with the healthier state of VT strain-infected placentas, the impairment of HV strain-infected placentas for control of immune responses, and the fact that low birth weight is observed for some babies born with congenital Chagas disease (36). Moreover, some of these pregnancy-related genes have been directly implicated in the regulation of microbial infections. Prolactin for instance, mainly modulated in placentas of the HV group, has been proposed as an immunomodulatory molecule that enhances *Toxocara canis* migration to the uterus and mammary glands, allowing vertical transmission (37). High prolactin serum levels have been associated with seronegativity to *Toxoplasma gondii* (62), which is capable of directly binding to the parasite and restricting intracellular growth and competent infection (reviewed in reference 38). Another pregnancy-related protein is pregnancy zone protein (PZP), an alpha macroglobulin that is highly expressed during late pregnancy and has the capacity to inhibit proteinases. PZP levels increase during the course of acute Chagas disease in children and reportedly interact with *T. cruzi*'s cruzipain, the parasite's main proteinase, inhibiting proteolytic capacity in order to minimize harm (39). In our experiments, PZP was downregulated in HV strain-infected placentas and unchanged in VT strain-infected placentas. The regulation patterns of these pregnancy-related genes obtained in this work support the differential modulation of our models, evidencing that the passage of VT strains does not interfere with healthy placental function, therefore allowing silent transplacental passage without major impact on pregnancy outcome.

It is worth mentioning that a group of genes related to membrane permeability and trafficking, encoding sodium channels and solute carriers, were upregulated in placentas of the VT strain-infected group but downregulated or unchanged in the HV group: these were *Scn9a*, *Slc22a13*, *Slc9b2*, *Slc6a1*, and *Slc17a2* (Fig. 8). Some of these genes have been previously associated with HIV infection and immunodeficiency and were modulated by hepatitis C virus (HCV) in favor of viral propagation and establishment of chronic infection (40). Specifically for *T. cruzi* placental infection, several other solute carriers were found to be upregulated in experimental infections (41). Parasites are known to regulate cell-cell adhesions as part of their pathogenic mechanisms, specifically by inhibiting gene expression and weakening unions (reviewed in reference 42).

A set of genes encoding key components of cell-cell and cell-matrix unions that confer tissue stability and prevent paracellular passage of pathogens (43) were upregulated in the VT strain-infected placentas and downregulated or unchanged in HV strain-infected placentas. In particular, *Cldn* and *Dsg* genes are noteworthy, since they are components of tight junctions and desmosomes respectively, both related to paracellular metabolite passage and conductance (Fig. 8). Additionally, *Ocln*, another component of tight junctions, was upregulated with TcKr transplacental passage. Taken together, these results indicate that in addition to an inflammatory response, the HV strain induces changes that favor paracellular passage, already described for several pathogens (42), including *T. cruzi* (44). In opposition, the VT strains induced junction-related genes, which should favor intracellular placental passage, contributing to immune escape. This type of strategy has been described for *Plasmodium*, where through a lysis-free mechanism of cell-to-cell passage called cell traversal, the parasite migrates through the tissue until it finds the proper microenvironment to settle and replicate (45). Transcriptomic results indicating tissue damage in placentas infected with the HV strains and, more importantly, maintenance of tissue integrity in placentas of the VT strain-infected groups are supported by the histopathological analysis summarized in Fig. 9. Finally, the results obtained for VT strains, involving low replication and high placental tropism coupled with immunological silence without tissue damage, suggest that this strategic mechanism is exploited by specialized *T. cruzi* isolates and should be further studied. Generally, transcriptomic analysis found in the literature describes placental responses that resemble our results regarding the HV strain, leading to the conclusion that parasite passage is more likely to be due to tissue damage and inflammation, which may lower the parasite load but also allow passage.

In summary, we identified the following two different vertical transmission strategies, which are represented schematically in Fig. 10.

1. Unspecific or nonpreferential tropism toward the placenta. These strains (HV) are vertically transmitted, inducing damage due to high virulence, evidenced by the proinflammatory placental response to infection. We think that this kind of virulence pattern does not necessarily correspond to what is seen in clinical cases of congenital Chagas disease, except for acute infection acquired during pregnancy, which presents higher vertical transmission frequencies (46, 47).

2. Specific or placentotropic. This is the case for VT isolates, which present mild immune response modulation and a characteristic profile or regulation involving small changes. The mechanism used by these strains constitutes a specialization to human infection and nonvectorial transmission of *T. cruzi*. It evidences an adaptive strategy to achieve vertical transmission without causing damage or illness that may jeopardize an individual host's health. We postulate that this is the most common strategy in congenital Chagas disease, supported by the observation that in most of these cases, the disease is diagnosed by screening during pregnancy, since no signs or symptoms present. In addition, previous results regarding exacerbated immune responses, like the ones we observed for the Garbani (HV) strain, are typical of virulent laboratory strains and biased because *in vitro* models fail to reproduce what happens in the maternal-fetal interphase.

## MATERIALS AND METHODS

**Clinical cases and diagnosis.** Case 1 (2013) was an asymptomatic 12-month-old baby born and living in a nonendemic area (Montevideo, Uruguay), with a positive mother (24 years old) who also lived in a nonendemic area and a positive grandmother and great grandmother living in a nonendemic area (Montevideo) and an endemic area (Salto, Uruguay), respectively. The weight and gestational age at birth were 3.400 kg and 38 weeks. Case 2 (2015) was an asymptomatic 1-month-old baby born in a nonendemic area (Montevideo, Uruguay) to a positive mother, also born and living in Montevideo, who had two other positive children. The weight and gestational age at birth were 3.910 kg and 39 weeks. Case 3 (2017) was a 3-month-old baby born asymptomatic in a nonendemic area who developed an atrioventricular blockage at 3 months of age; the mother was positive. The weight and gestational age at birth were 3.320 kg and 41 weeks.

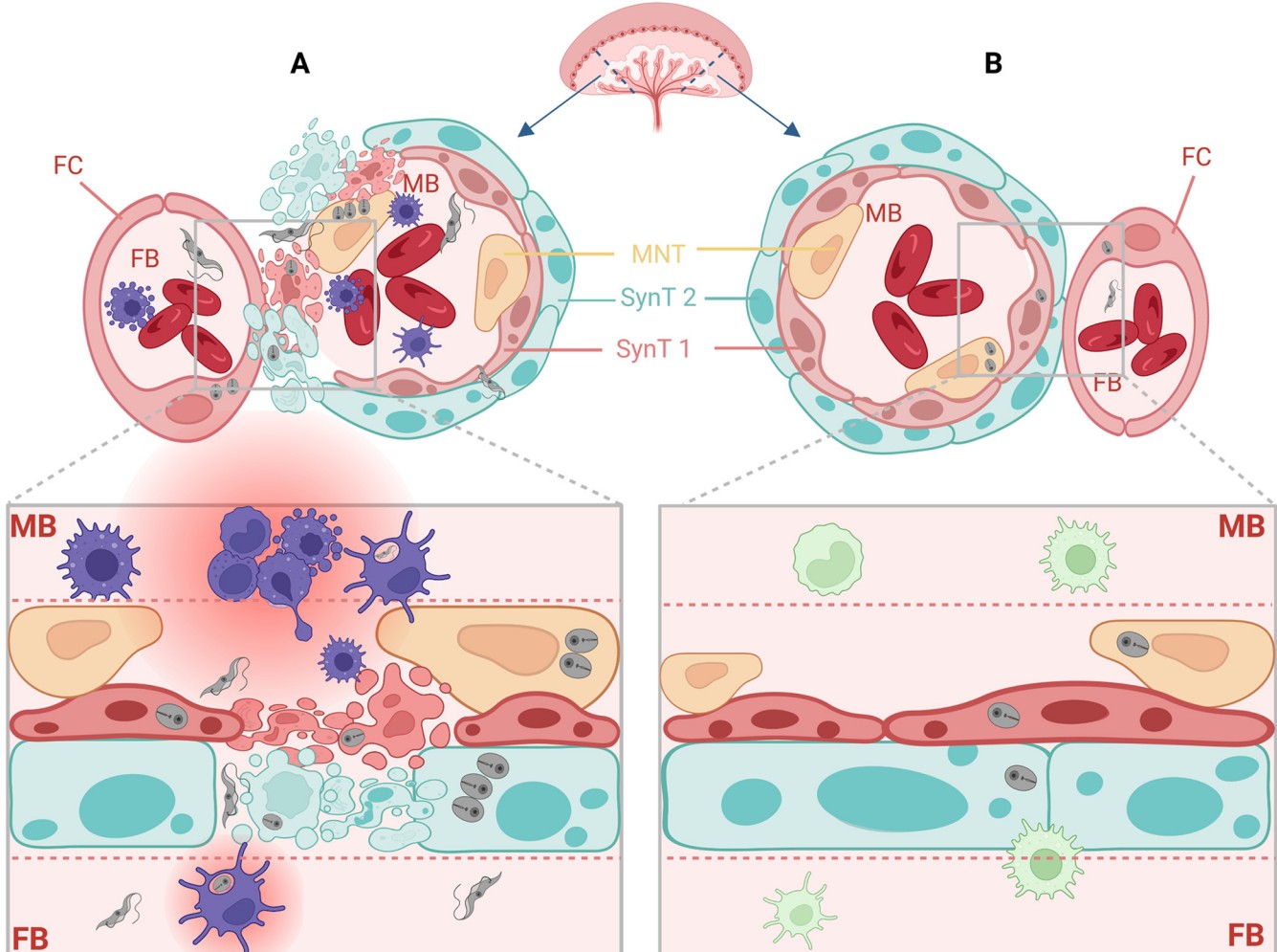

**FIG 10** Two vertical transmission strategies at the murine maternal-fetal interphase. (A) As observed for the highly virulent strain Garbani, *T. cruzi* is vertically transmitted, unspecifically and without preferential tropism toward the placenta, inducing tissue damage and a strong proinflammatory placental response. (B) As observed for VT isolates, *T. cruzi* is vertically transmitted, specifically and with a placentotrophic strategy, in a silent manner, without causing damage and without inducing a proinflammatory response. Figure created with biorender.com.

All of the mothers and babies were born in nonendemic areas. Xenodiagnosis (5 inbred stage IV nymphs of *Triatoma infestans*) was performed in babies, and after bites, nymphs' feces were monitored at 20, 40, and 60 days postdiagnosis. The procedure was repeated every 6 months until the baby was either positive or 18 months old. Xenodiagnosis was considered positive when *T. cruzi*-compatible trypomastigotes were observed in at least one nymph; positive patients were started on benznidazol treatment. Congenital infection was inferred due to patients residing in nonendemic areas and never having traveled or received blood transfusions. A summary of the characteristics of each clinical case is given in Table 1.

**Animals.** Inbred BALB/cJ and Nu/J 8-week-old female mice (Jackson Laboratory, Bar Harbor, USA) were housed under specific-pathogen-free conditions in individually ventilated cages (IVCs; Tecniplast, Milan, Italy) at the Animal Facility of the Unit of Biotechnology of Laboratory Animals. During infection, mice were assigned to groups of no more than six and housed in a biosafety level 2 (BSL2) ventilated rack with HEPA filters (IsoCage N; Tecniplast). All infection procedures were performed under a BSL2 laminar flow hood to comply with biosafety institutional rules. Animals received autoclaved fresh water and a standard autoclaved mouse diet *ad libitum* (Labdiet 5K67; PMI Nutrition, USA).

**Strain isolation and maintenance.** Feces of positive nymphs were resuspended in salinated water and inoculated into BALB/cJ mice (Jax strain No. 000651) via the intraperitoneal (i.p.) route. Parasitemia was assessed by microscopic observation of blood samples obtained by submandibular puncture every 3 days. When parasitemia was detected, an immunosuppression protocol with cyclophosphamide (Filaxis) was applied at 40 mg/kg of body weight/day for 5 days via the i.p. route. Afterwards, complete blood was used to inoculate three Nu/J mice (Jax strain no. 002019) to amplify the parasite load and create parasite stocks for further use in the experiments. Nu/J mice were infected through the i.p. route with $2 \times 10^5$ parasites, and parasitemia was monitored 2 to 3 times a week by blood parasite counting obtained by puncture of the submandibular vein until a parasitemia peak of $5 \times 10^6$ parasites/mL was reached. Thereafter, mice were deeply anesthetized via the i.p. route with a mixture of ketamine

**TABLE 1** Summary of characteristics of the clinical cases from which VT strains were isolated

| | Baby | | | | | Mother | | | | Grandmother | | | Great grandmother | | | Infected siblings |
|---|---|---|---|---|---|---|---|---|---|---|---|---|---|---|---|---|
| Case | D | S/A | Birthplace | Wt (kg) | GA (wk) | D | S/A | Birthplace | Age (yr) | D | S/A | Birthplace | D | S/A | Birthplace | |
| 1 (TcLu) | + | A | NE | 3.4 | 38 | + | A | NE | 24 | + | U | NE | + | U | E | No |
| 2 (TcGi) | + | A | NE | 3.9 | 39 | + | A | NE | U | U | U | U | U | U | U | Yes |
| 3 (TcKr) | + | S | NE | 3.3 | 41 | + | A | U | U | U | U | U | U | U | U | No |

*a*D, diagnosis, via positive or negative xenodiagnosis of *T. cruzi* infection in the case of the babies and serology in the case of the adults; S, symptomatic; A, asymptomatic; NE, nonendemic area; E, endemic area; GA, gestational age at birth; Age, age of mother at baby's birth; U, unknown.

(100 mg/kg, vetanarcol; König, Buenos Aires, Argentina) and xylazine (10 mg/kg, seton; Calier, Barcelona, Spain) for final bleeding. Blood was collected by cardiac puncture, and mice were immediately euthanized by cervical dislocation. Total parasites were purified by differential centrifugation in a Percoll gradient as described elsewhere (48) and counted to be used in further experiments. The first parasitemia peak in nude mice was considered passage 1 (P1), and experiments were always performed with P1 to P5 parasites to conserve biological features. P1 to P5 *T. cruzi* VT isolates from case 1 (TcLu), case 2 (TcGi), and case 3 (TcKr) and trypomastigotes from the Dm28c (DTU TcI) and Garbani (DTU TcVI) strains were purified every time from nude mouse blood. Fresh parasites were used in all the experiments conducted in this study.

**Molecular typing.** Purified DNA from all strains was obtained from trypomastigotes and was used to identify the discrete typing unit (DTU) based on specific PCR products, including the intergenic region of spliced leader genes (SL-IR), the 24S$\alpha$ ribosomal DNA subunit (rDNA 24S$\alpha$), and the A10 fragment previously described (49). PCR products were seeded in 5% MetaPhor agarose gels, and amplification products were compared to reference strains. PCR products were purified and verified by Sanger sequencing. DTU determination was afterwards confirmed by Alejandro Shijman (Instituto de Investigaciones en Ingeniería Genética y Biología Molecular Dr. Héctor N. Torres INGEBI-CONICET, Buenos Aires, Argentina) using a multiplex real-time PCR developed by that group (50).

**Epimastigogenesis.** Trypomastigotes purified from mouse blood were incubated in phosphate-buffered saline (PBS; pH 6) at 37°C for 4 h before epimastigogenesis in liver infusion tryptose (LIT) as previously described (15). Different time points were set for collection to count epimastigote forms and for immunofluorescence staining. Epimastigotes were quantified with the following equation: % epimastigotes = (number of epimastigotes/total number of parasites) $\times$ 100.

**Cell culture.** HFF-1 cells (human foreskin fibroblasts; SCRC-1041) were purchased from ATCC, and Swan 71 cells (51) were kindly donated by Gil Mor (C. S. Mott Center for Human Growth and Development, Department of Obstetrics and Gynecology, Wayne State University). Both cell lines were grown in Dulbecco's modified Eagle's Medium (DMEM) (Life Technologies, Grand Island, NY, USA) supplemented with 10% fetal bovine serum (FBS) at 37°C in a 5% $CO_2$ humidified atmosphere. The epimastigote forms used were grown in LIT supplemented with 10% FBS at 28°C.

**Determination of infectivity index in mammalian cells.** Nude mouse trypomastigotes derived from all strains were purified, counted, and incubated with HFF-1 and Swan 71 semiconfluent cell monolayers for 2 h at a multiplicity of infection of 5 parasites to 1 host cell. Afterwards, noninternalized trypomastigotes were washed 5 times with PBS and cells were incubated for another 48 h before immunofluorescence microscopy. After 48 h, coverslips were washed with PBS, fixed with 95% (vol/vol) ethanol, and stained with Fluoroshield with DAPI (4′,6-diamidino-2-phenylindole; Sigma, USA). Infectivity was evaluated considering invasion and replication capacity by counting infected cells and parasites per infected cell in the infection photos obtained from each experimental condition. For each replicate, a total of at least 500 cells were counted and the results are expressed in graphs as the mean values and standard errors (SE) of 3 independent experiments. The infection index was calculated as follows: infection index = (number of amastigotes $\times$ number of infected cells)/total number of cells.

**Indirect immunofluorescence.** Infected cells plated on coverslips or parasite pellets were washed with PBS and fixed with PBS–4% paraformaldehyde, permeabilized for 5 min with 0.1% Triton X-100, and blocked for 30 min with 5% bovine serum albumin (BSA) (all reagents purchased from Sigma). Cells from infectivity and replications assays were incubated for 1 h at room temperature with anti-cytosolic tryparedoxin peroxidase (cTXNPx) primary antibody (52) and for 30 min with Alexa Fluor 488 anti-rabbit IgG secondary antibody (Invitrogen, Eugene, OR). Parasites from the epimastigogenesis experiment were attached to poly-L-lysine-coated slides and treated as monolayers, except for those incubated with a monoclonal primary antibody (MAb25, kindly provided by Sergio Schenkman; 1:1,000 dilution) that binds to the *T. cruzi* flagellar calcium-binding protein (53). All antibodies were diluted in 1% BSA/PBS, and coverslips and parasites were mounted with ProLong antifade reagent.

**Mouse infection, monitoring, survival, and parasitemia curves.** For mouse survival and parasitemia curves, groups of six 6- to 8-week-old BALB/cJ females were inoculated with Nu/J-derived trypomastigotes and parasitemia was monitored as described above. An uninfected control group was included in every experiment. The initial trypomastigote dosages were $5 \times 10^4$ and $1 \times 10^4$ for the survival and parasitemia curves, except for the Garbani strain, for which the dosages were $10^4$ and $10^3$ due to its virulence. On postinfection (p.i.) day 35, when parasites were not visually detected in blood, mice were anesthetized by intraperitoneal inoculation of a mixture of ketamine (100 mg/kg) and xylazine (10 mg/kg). Blood was collected by cardiac puncture, and mice were euthanized by cervical dislocation. Blood was

placed into dry or heparinized tubes for serum collection or molecular testing, respectively. Organs (heart, gut, spleen, and uterus) were perfused with ice-cold PBS, and except for a piece of the spleen that was dissected and immediately stored at −80°C in TRIzol reagent (Invitrogen) until RNA extraction, the rest of the spleen and organs were dry-frozen at −80°C for further DNA extraction.

**Evaluation of mating, pregnancy, implantation, and vertical transmission.** For vertical transmission experiments, 8-week-old BALB/cJ female mice were randomly separated into six groups (six mice per group) and infected with the different strains. An uninfected control group was also included. The initial inoculums were $5 \times 10^4$ for VT strains, $1 \times 10^4$ for Dm28c, and $1 \times 10^3$ for Garbani. Injection and monitoring were performed as described above. To evaluate the reproductive parameters and vertical transmission, 30 days after infection, infected females were placed for 2 days in cages with dirty bedding from males (Whitten effect) to synchronize their estrous cycles. Afterwards, mating was performed by adding one proven male in a cage with 3 females for 5 days. Females were checked every day for the presence of a vaginal plug (VP), and the average of VP-positive females registered during those 5 days was used to calculate mating percentage as described below. All females were anesthetized and euthanized on day 20 of gestation, and viable fetuses and resorptions were recorded to measure pregnancy and implantation rates. Maternal blood, heart, gut, spleen, and uterus and fetal, placental, and resorption tissues were collected and processed according to the experiment procedure for each case. Rates were quantified with the following equations: % mating = (number of females with VP/total number of females) × 100; % pregnancy = (number of pregnant females/total number of females) × 100; % embryo resorption = (number of resorptions/total number of viable fetuses and resorptions) × 100; and vertical transmission rate = (number of positive viable fetuses/total number of pregnant females) × 100.

***T. cruzi* DNA detection by qPCR.** Blood DNA was extracted using the Quick-DNA miniprep plus kit (catalog number D4069; Zymo Research), and organ DNA using the Quick-DNA tissue/insect plus kit (catalog number D6016; Zymo Research) according to the manufacturer's recommendations. Tissue disruption was achieved using the Bullet Blender tissue homogenizer (Next Advanced) and the bead kit. Total DNA was quantified and diluted to 100 ng/$\mu$L, and 1 $\mu$L of the sample was amplified using a kinetoplast DNA (kDNA) TaqMan probe (54) with an Integrated Data Technologies (IDT) mixture in a final volume of 10 $\mu$L, and the relative expression was calculated by the cycle threshold ($\Delta\Delta C_T$) method and normalized against the mouse *gapdh* (glyceraldehyde-3-phosphate dehydrogenase) housekeeping gene. Samples were analyzed in duplicate using the QuantStudio 3 thermocycler (Applied Biosystems, Thermo Fisher Scientific). Primers and probe sequences are detailed in Table S1. On the plates, reaction controls with water, positive controls with *T. cruzi* genomic DNA (gDNA), and negative controls with DNA from uninfected mice were included. The latter was used to set up a negative-cutoff line for PCR results in infected organs.

**Cytokine expression.** Mice infected with *T. cruzi* were euthanized at 30 and 60 dpi, and the spleens were removed and subjected to RNA extraction with TRI Reagent (Invitrogen) and the Direct-zol RNA miniprep plus kit (catalog number R2072; Zymo Research) according to the manufacturer's instructions. cDNA was synthesized from 500 ng of RNA using SuperScript II reverse transcriptase (Invitrogen) using oligo(dT). Reactions were performed in a final volume of 10 $\mu$L using SYBR green (KAPA SYBR fast universal 2× quantitative reverse transcription-PCR [qRT-PCR] master mix), a final primer concentration of 200 nM, and 1 $\mu$L of a 1/5 dilution of the cDNA. Each reaction was performed in duplicate in the QuantStudio 3 thermocycler (Applied Biosystems, Thermo Fisher Scientific). Melt curves were generated to check specific amplification products, and the $C_T$ values obtained for each gene were normalized against the expression of the *Mus musculus gapdh* gene. The $\Delta C_T$ was determined to calculate the relative expression for each gene in every sample by the $\Delta\Delta C_T$ method. The primer sequences used to amplify the different cytokines are listed in Table S1.

**Transcriptomics.** For the RNA-seq experiment, three placentas for each condition (biological replicates) were selected and processed to extract total RNA. Briefly, placentas stored at −80°C were disrupted in TRI Reagent (Thermo Fisher) in a Bullet Blender (Next Advance) with 1.6-mm stainless steel beads (SKU SSB16; Next Advance) for 10 min at maximum velocity. Once the tissue was disrupted, 200 $\mu$L of chloroform for 1 mL of TRI Reagent was added and the mixture homogenized and centrifuged for 15 min at 15,000 relative centrifugal force (rcf). The aqueous phase was recovered and processed with a Direct-zol RNA kit (Zymo Research, USA) according to the manufacturer's instructions. RNA quality was determined using the Bioanalyzer RNA 6000 nanokit (Agilent, USA) in a Bioanalyzer 2100 (Agilent, USA). An RNA integrity number (RIN) higher than 8 was considered acceptable to continue to library construction and sequencing. Library construction and sequencing were performed by Macrogen (Korea). For library construction, TruSeq stranded total RNA with Ribo-Zero Gold (Illumina, USA) was used. Sequencing was performed on the Illumina NovaSeq6000 platform. Paired-end reads of 100 bp were obtained. More than 60 million reads were obtained for each sample (Table S2). Raw data were deposited in NCBI (SRA) under accession number PRJNA820598. The sequences obtained were quality checked using FastQC (55). Reads were mapped against Mouse Genome version 39 (GRCm39.primary_assembly genome download from Ensembl) with a STAR package (56) using annotation file M26 released by the Gencode project. Reads counts were performed using FeatureCounts (57), and mapping statistics and STAR and FeatureCounts parameters are included in Table S2. FeatureCounts output was processed in RStudio using the DESeq2 package (58) to obtain differentially expressed genes (DEG) using $|\log_2 FC| > 1.5$ and a Benjamini-Hochberg-adjusted *P* value of <0.05 (or <0.01 when indicated) to perform downstream analysis. A table of DEGs was generated for each condition compared to the control group (uninfected animals) (Table S3). DEG cluster analysis was performed using the pheatmap package in RStudio software or Graph Prism version 9.0. Gene Ontology analyses were performed in Metascape (59).

**Histopathology.** Immediately after the autopsy, whole placentas were fixed in 10% neutral buffered formalin (pH 7.4) for further processing. After 5 days of fixation, all the placentas were transversally

hemisected and processed simultaneously in an automatic tissue processor (SLEE MTP carousel tissue processor; SLEE medical, Germany), embedded in paraffin (SLEE MPS/P1 dispensing module; Germany), sectioned at 5 $\mu$m (Leitz 512 rotatory microtome, Germany), deparaffined, rehydrated, stained with hematoxylin and eosin (H&E) (SLEE MSM Carousel slide stainer, Germany), rinsed, dehydrated, cleared, and mounted according to the method of Sala et al. (60). For histological evaluation, the specimens were examined under a light microscope (Olympus BX41) at 10× magnification, considering the selected regions of the maternal decidua basalis, trophoblastic giant cell zone, spongiotrophoblast zone, and labyrinth zone, according to Georgiades et al. (61). Each specimen was evaluated by two different pathologists (B.V. and J.M.V.) to establish a histopathological score in each case.

**Ethical statement.** For all samples, we obtained informed consent from the patients and informed permission for the child, and we adhered to ethical standards in study design and execution. The isolation experiments were done following protocols preapproved by the institutional ethics committee (CEUA protocols 018–22 and 019–22) at the Institut Pasteur de Montevideo and in accordance with National Law 18.611 (https://www.impo.com.uy/bases/leyes/18611-2009/5).

**Data availability.** Raw data were deposited in NCBI (SRA) under accession number PRJNA820598.

## SUPPLEMENTAL MATERIAL

Supplemental material is available online only.
**SUPPLEMENTAL FILE 1**, PDF file, 3.5 MB.

## ACKNOWLEDGMENTS

We thank Estela Bevilacqua for critical reading of the manuscript, Alejandro Schijman for confirming our DTU results, Gil Mor for kindly sharing the Swan 71 cell line, Miriam Postan for kindly providing the Garbani strain, Guzmán Alvarez for benznidazol and nifurtimox drugs, Sergio Schenkman for Mab25 antibody, and Grazzia Rey for helpful advice.

This work was supported by ERANET-LAC project ERANet17/HLH0142 (to C.R.). P.F.-T. was a PEDECIBA student and received grants from Agencia Nacional de Investigación e Innovación (ANII, Uruguay) and Comisión Académica de Posgrado (Universidad de la República, Uruguay). P.F.-T., G.G., G.L., A.C., J.M.V., and C.R. are members of the Sistema Nacional de Investigadores (SNI, ANII). G.G., M.C., P.F.T., and C.R. are PEDECIBA researchers.

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
