## [Reviewer comments · Microbiology Spectrum]

Microbiology Spectrum

Trypanosoma cruzi isolates naturally adapted to congenital transmission display a unique strategy of transplacental passage

Paula Faral-Tello, Gonzalo Greif, Selva Romero, Andres Cabrera, Cristina Oviedo, Telma Gonzalez, Gabriela Libisch, Ana Paula Arévalo, Belen Varela, José Verdes, Martina Crispo, Yester Basmadjian, and Carlos Robello

Corresponding Author(s): Carlos Robello, Institut Pasteur de Montevideo

Review Timeline:

Submission Date:	July 13, 2022
Editorial Decision:	September 6, 2022
Revision Received:	December 21, 2022
Accepted:	January 12, 2023

Editor: Denis Sereno

Reviewer(s): Disclosure of reviewer identity is with reference to reviewer comments included in decision letter(s). The following individuals involved in review of your submission have agreed to reveal their identity: Ulrike Kemmerling (Reviewer #1)

Transaction Report:

DOI: <https://doi.org/10.1128/spectrum.02504-22>

September 6, 2022

Prof. Carlos Robello
Institut Pasteur de Montevideo
Unidad de Biología Molecular
Mataojo 2020
Montevideo 11400
Uruguay

Re: Spectrum02504-22 - CORRECTION (Trypanosoma cruzi isolates naturally adapted to congenital transmission display a unique strategy of transplacental passage)

Dear Prof. Carlos Robello:

Substantial modifications to the manuscript must be performed before being considered for publication in Microbiology Spectrum. 1)The references list is lacking in the paper, 2) "Informed consent" of the patients and "Informed permission" for the child are lacking, 3) Adherence of the clinical study to ethical standards in study design and execution, monitored by an Institutional Review Bord is not quoted in the papaer.

Link Not Available

Sincerely,

Denis Sereno

Journals Department
Reviewer comments:

Reviewer #1 (Comments for the Author):

The manuscript "Trypanosoma cruzi isolates naturally adapted to congenital transmission display a unique strategy of transplacental passage" is an interesting study about the mechanisms of congenital transmission.

However, there are certain issues that must be addressed before accepting the manuscript:

- 1) the histopathological analysis is made with bad processed samples. The tissue have not been fixed and dehydrated in the correct way and some of the damage described in the text could be due to the wrong processing. Intracellular vesiculations is normally due to an excess of dehydrtation, separation of the tissue is due to wrong fixation. Better images must be provided.
- 2) Human and mouse placenta are structurally different, this should be mentioned and discussed in the manuscript.

Staff Comments:

Preparing Revision Guidelines

Please return the manuscript within 60 days; if you cannot complete the modification within this time period, please contact me. If you do not wish to modify the manuscript and prefer to submit it to another journal, please notify me of your decision immediately so that the manuscript may be formally withdrawn from consideration by Microbiology Spectrum.

Spectrum02504-22 - CORRECTIONR1

Response to Reviewers

The manuscript "*Trypanosoma cruzi* isolates naturally adapted to congenital transmission display a unique strategy of transplacental passage" is an interesting study about the mechanisms of congenital transmission.

However, there are certain issues that must be addressed before accepting the manuscript:

1) the histopathological analysis is made with bad processed samples. The tissue have not been fixed and dehydrated in the correct way and some of the damage described in the text could be due to the wrong processing. Intracellular vesiculations is normally due to an excess of dehydration, separation of the tissue is due to wrong fixation. Better images must be provided.

Response:

We thanks this comment. Taking it into account, we have repeated the experiments. We used fresh placentas that were immediately fixed and processed, and we incorporated an automatic method in order to process all the samples at the same time and conditions, in order to follow the suggestion of the reviewer. We used an SLEE MTP® Carousel tissue processor, and analyzed the new samples. The new Figure 9 is included, and changes in the section 2.14 include these modifications. In addition, we made changes in the legend of Figure 9.

2) Human and mouse placenta are structurally different, this should be mentioned and discussed in the manuscript.

I totally agree with this comment. In the discussion, we added the sentence, "Although there is a difference between the mouse and human placenta with respect to the depth of trophoblast invasion into decidual tissue, both are hemochorial placentas in which the fetal trophoblast is in direct contact with maternal blood."

January 3, 2023

Prof. Carlos Robello
Institut Pasteur de Montevideo
Unidad de Biología Molecular
Mataojo 2020
Montevideo 11400
Uruguay

Re: Spectrum02504-22 - CORRECTIONR1 (Trypanosoma cruzi isolates naturally adapted to congenital transmission display a unique strategy of transplacental passage)

Dear Prof. Carlos Robello:

Thank you for considering the reviewer comments in the new version of your paper as well as the ethical concerns.

Your manuscript has been accepted, and I am forwarding it to the ASM Journals Department for publication. You will be notified when your proofs are ready to be viewed.

Sincerely,

Denis Sereno
Editor, Microbiology Spectrum
